# A multi-ethnic meta-analysis identifies novel genes, including *ACSL5*, associated with amyotrophic lateral sclerosis

Ryoichi Nakamura ⬤ et al.[#]

Amyotrophic lateral sclerosis (ALS) is a devastating progressive motor neuron disease that affects people of all ethnicities. Approximately 90% of ALS cases are sporadic and thought to have multifactorial pathogenesis. To understand the genetics of sporadic ALS, we conducted a genome-wide association study using 1,173 sporadic ALS cases and 8,925 controls in a Japanese population. A combined meta-analysis of our Japanese cohort with individuals of European ancestry revealed a significant association at the *ACSL5* locus (top SNP $p = 2.97 \times 10^{-8}$). We validated the association with *ACSL5* in a replication study with a Chinese population and an independent Japanese population (1941 ALS cases, 3821 controls; top SNP $p = 1.82 \times 10^{-4}$). In the combined meta-analysis, the intronic *ACSL5* SNP rs3736947 showed the strongest association ($p = 7.81 \times 10^{-11}$). Using a gene-based analysis of the full multi-ethnic dataset, we uncovered additional genes significantly associated with ALS: *ERGIC1*, *RAPGEF5*, *FNBP1*, and *ATXN3*. These results advance our understanding of the genetic basis of sporadic ALS.

---

[#]A list of authors and their affiliations appears at the end of the paper.

Amyotrophic lateral sclerosis (ALS) is the most common motor neuron disease characterised by progressive skeletal muscle atrophies that lead to death mostly within 3–5 years from disease onset[1]. Mutations in more than 30 genes have been identified to cause ALS, including *SOD1, TARDBP, FUS*, and *C9orf72*. Approximately 50–70% of ALS patients with a family history of the disease can be attributed to mutations in these genes; however, these mutations are found in only 3% of the Japanese patients with sporadic ALS[2]. A twin study estimated the heritability of sporadic ALS to be 61%[3]; therefore, sporadic ALS is thought to be a multifactorial disease to which multiple genetic and environmental factors contribute. Previous genome-wide association studies (GWAS) reported six common loci showing genome-wide significant associations with ALS, which only explained 0.2% of its heritability[4]. To elucidate the pathophysiology of sporadic ALS and to develop appropriate therapies, the genetic background of sporadic ALS needs to be understood more clearly.

Assuming that common causal variants in different populations are important in investigating the pathophysiology of sporadic ALS, cross-ethnic meta-analysis of GWAS would have valuable implications. In addition, cross-ethnic GWAS has an advantage for fine-mapping because the linkage disequilibrium patterns that differ across populations can improve the resolution[5]. Thus, we report analyses of novel genome-wide association study data of 1173 sporadic ALS cases and 8925 normal controls in a Japanese population and meta-analysis with the largest ALS study in a European population[6]. We also validate the candidate region with a combined replication study using 707 other ALS cases and 971 controls from a Japanese population and a Chinese dataset[7]. Using a gene-based analysis, we identify four additional genes significantly associated with ALS. The discovery of novel risk genes advances our understanding of ALS aetiology.

## Results

**Genome-wide association analysis**. To identify new susceptibility loci operating in sporadic ALS, we conducted a GWAS in a Japanese sample of 1173 sporadic ALS cases and 8925 controls (Supplementary Figs. 1, 2, and 3). The 56 SNPs passed the condition with $p < 5.0 \times 10^{-6}$ (Supplementary Data 1), while no individual SNPs passed the genome-wide significant $p$-value threshold of $5.0 \times 10^{-8}$. The data were then incorporated into a meta-analysis with a large-scale GWAS involving 20,806 patients diagnosed with ALS and 53,439 control subjects of European ancestry[6]. The Manhattan plot in Fig. 1 shows four identified loci in the European population that achieved genome-wide significance (Fig. 1a; loci include *GPX3-TNIP1, C9orf72, TBK1*, and *UNC13A*; Fig. 1b)[4,6,7]. In addition, we found the novel genome-wide statistical signature for SNPs in linkage disequilibrium on chromosome 10q25.2. This region spans several hundred kilobases and encompasses *ACSL5* (Supplementary Data 2). Table 1 lists the top three SNPs displaying $p < 5.0 \times 10^{-8}$ in the *ACSL5* gene body region: rs58854276 ($p = 2.97 \times 10^{-8}$; odds ratio (OR) $= 1.080$; 95%CI $= 1.065$–$1.095$), rs11195948 ($p = 3.99 \times 10^{-8}$; OR $= 1.079$; 95%CI $= 1.064$–$1.094$), and rs3736947 ($p = 3.61 \times 10^{-8}$; OR $= 1.080$; 95%CI $= 1.065$–$1.095$). The top three SNPs in *ACSL5* are all intronic variants. Figure 2 shows the regional mapping of this region.

**Replication analysis**. The association result was replicated in the combined meta-analysis with the Chinese dataset (1234 ALS cases and 2850 controls)[7] and an independent new Japanese dataset (707 ALS cases and 971 controls). Among the former top three SNPs, two SNPs—rs11195948 and rs3736947—were validated (Table 1, Fig. 3, Supplementary Fig. 4). The $p$-value of rs11195948

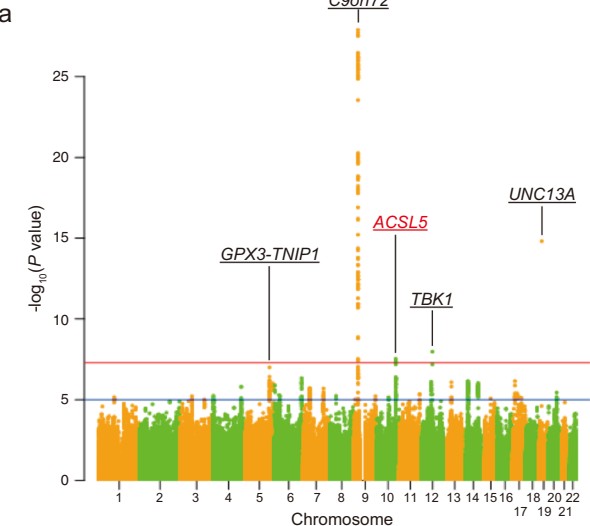

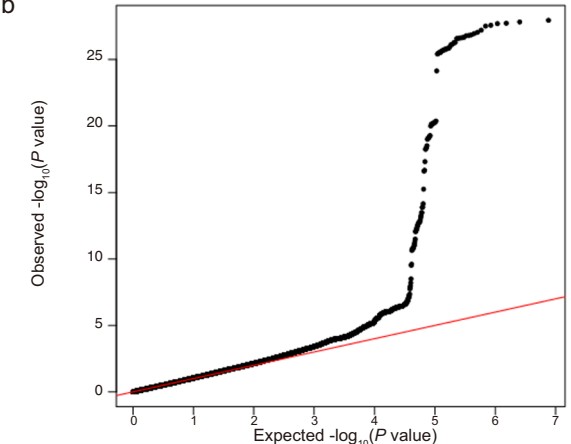

**Fig. 1 Meta-analysis between European and Japanese genome-wide association studies (GWAS) revealed a novel locus. a** Manhattan plot of the meta-analysis between European and Japanese GWAS. The novel locus *ACSL5* is shown in red. Loci identified in other studies are presented with lines along the gene names, i.e. *GPX3-TNIP1, C9orf72, TBK1*, and *UNC13A*. The red horizontal line indicates the genome-wide significance threshold of $p = 5 \times 10^{-8}$. The blue line indicates the suggestive threshold of $p = 5 \times 10^{-6}$. The $p$-values of Japanese GWAS were calculated from the logistic regression model. **b** Q-Q plot of the meta-analysis between European and Japanese GWAS.

was the most significant ($p = 1.82 \times 10^{-4}$; OR $= 1.136$; 95%CI $= 1.098$–$1.176$). The tendency of the ratio in the Chinese sample was in the same direction ($p = 2.09 \times 10^{-4}$; OR $= 1.199$; 95%CI $= 1.103$–$1.290$, Table 1, Fig. 3) as that of the European and initial Japanese samples in the discovery dataset. However, the result of the Japanese replication dataset was not conclusive ($p = 0.554$; OR $= 0.960$; 95%CI $= 0.838$–$1.100$, Fig. 3, Supplementary Fig. 4). For the combined dataset of the discovery and replication studies, the most significant SNP was rs3736947 ($p = 7.81 \times 10^{-11}$; OR $= 1.087$; 95%CI $= 1.073$–$1.101$, Table 1, Fig. 3).

**Multi-ethnic meta-analysis**. To maximise the power to detect genes related to ALS, we conducted the largest meta-analysis, with 23,213 cases and 71,579 controls from Japanese, European, and Chinese populations (Fig. 4a, b, and Supplementary Data 3). In the meta-analysis, the SNPs in *ACSL5* also reached genome-wide significance: rs58854276 ($p = 6.23 \times 10^{-10}$), rs11195948 ($p$

**Table 1 Top three SNPs displaying $p < 5.0 \times 10^{-8}$ in the *ACSL5* gene body region by GWAS of the meta-analysis between European and Japanese (JaCALS) cohorts.**

| SNP | Chr | Position | Cohort | Population | N cases | N control | EA | OA | OR | 95% CI | P-value | Direction | Imputed/ Direct | Info Score or Call Rate |
|---|---|---|---|---|---|---|---|---|---|---|---|---|---|---|
| rs58854276 | 10 | 114145044 | Discovery Cohort | European | 20,806 | 59,804 | A | G | 1.073 | 1.045–1.101 | $1.06 \times 10^{-6}$ | | NA | NA |
| | | | | Japanese | 1173 | 8925 | A | G | 1.183 | 1.066–1.313 | $1.62 \times 10^{-3}$ | | Imputed | 0.995 |
| | | | | Meta-analysis (European + Japanese) | 21,979 | 68,729 | A | G | 1.080 | 1.065–1.095 | $2.97 \times 10^{-8}$ | ++ | | |
| | | | Replication Cohort | Chinese | 1234 | 2850 | A | G | 1.204 | 1.079–1.361 | $1.46 \times 10^{-4}$ | | Imputed | >0.8 |
| | | | | Japanese 2nd | 707 | 971 | G | A | 1.021 | 0.891–1.171 | $7.62 \times 10^{-1}$ | | Direct | 0.995 |
| | | | | Meta-analysis (Chinese + Japanese 2nd) | 1941 | 3821 | A | G | 1.111 | 1.064–1.161 | $1.56 \times 10^{-2}$ | +− | | |
| | | | Meta-analysis (all) | European + Japanese + Chinese + Japanese 2nd | 23,920 | 72,550 | A | G | 1.083 | 1.069–1.097 | $1.79 \times 10^{-9}$ | +++− | | |
| rs11195948 | 10 | 114163515 | Discovery Cohort | European | 20,806 | 59,804 | C | T | 1.073 | 1.045–1.101 | $9.14 \times 10^{-7}$ | | NA | NA |
| | | | | Japanese | 1173 | 8925 | C | T | 1.163 | 1.049–1.289 | $4.18 \times 10^{-3}$ | | Imputed | 0.999 |
| | | | | Meta-analysis (European + Japanese) | 21,979 | 68,729 | C | T | 1.079 | 1.064–1.094 | $3.99 \times 10^{-8}$ | ++ | | |
| | | | Replication Cohort | Chinese | 1234 | 2850 | C | T | 1.199 | 1.103–1.290 | $2.09 \times 10^{-4}$ | | Imputed | >0.8 |
| | | | | Japanese 2nd | 707 | 971 | T | C | 1.042 | 0.909–1.194 | $5.54 \times 10^{-1}$ | | Direct | 0.996 |
| | | | | Meta-analysis (Chinese + Japanese 2nd) | 1941 | 3821 | C | T | 1.136 | 1.098–1.176 | $1.82 \times 10^{-4}$ | +− | | |
| | | | Meta-analysis (all) | European + Japanese + Chinese + Japanese 2nd | 23,920 | 72,550 | C | T | 1.087 | 1.073–1.101 | $8.20 \times 10^{-11}$ | +++− | | |
| rs3736947 | 10 | 114169190 | Discovery Cohort | European | 20,806 | 59,804 | C | A | 1.073 | 1.045–1.102 | $9.27 \times 10^{-7}$ | | NA | NA |
| | | | | Japanese | 1173 | 8925 | C | A | 1.163 | 1.050–1.290 | $3.97 \times 10^{-3}$ | | Direct | 1.00 |
| | | | | Meta-analysis (European + Japanese) | 21,979 | 68,729 | C | A | 1.080 | 1.065–1.095 | $3.61 \times 10^{-8}$ | ++ | | |
| | | | Replication Cohort | Chinese | 1,234 | 2,850 | C | A | 1.197 | 1.101–1.290 | $2.41 \times 10^{-4}$ | | Direct | >0.99 |
| | | | | Japanese 2nd | 707 | 971 | A | C | 1.038 | 0.906–1.190 | $5.90 \times 10^{-1}$ | | Direct | 0.997 |
| | | | | Meta-analysis (Chinese + Japanese 2nd) | 1941 | 3821 | C | A | 1.136 | 1.098–1.175 | $1.97 \times 10^{-4}$ | +− | | |
| | | | Meta-analysis (all) | European + Japanese + Chinese + Japanese 2nd | 23,920 | 72,550 | C | A | 1.087 | 1.073–1.101 | $7.81 \times 10^{-11}$ | +++− | | |

Position is based on Human Genome Assembly build 37.
*Chr* chromosome, *EA* effect allele, *OA* other allele, *OR* odds ratio, *CI* confidence interval, *NA* not available.

$= 1.79 \times 10^{-11}$), and rs3736947 ($p = 1.84 \times 10^{-11}$). In addition, a novel locus in the long non-coding RNA *TSBP1-AS1* reached genome-wide significance (rs140736091, $p = 1.36 \times 10^{-8}$), although this locus requires a replication study.

**Functional analysis of *ACSL5*.** A multi-tissue eQTL database GTEx v8 (https://www.gtexportal.org/home/)[8] revealed a significant relationship between rs3736947 and *ACSL5* expression. The risk allele (C) is associated with higher *ACSL5* expression than the non-risk allele (A) ($p = 6.5 \times 10^{-49}$, normalised effect size $= -0.39$ in the whole-blood dataset). To confirm the association, we conducted gene expression analysis for *ACSL5* by quantitative real-time PCR using lymphoblastoid B cell lines (LCLs) from age-matched and sex-matched Japanese patients with ALS with each genotype. The expression of *ACSL5* was significantly higher in the LCLs of patients with the CC ($n = 20$) and AC ($n = 20$) genotype of rs3736947 than in those with the AA genotype ($n = 20$) (CC vs AA, $p = 0.0035$, AC vs AA, $p = 0.0166$, Kruskal–Wallis test followed by Steel–Dwass test, Fig. 5, Supplementary Data 4).

**Gene-based association analysis.** We performed gene-based association analysis in the largest meta-analysis set, with 23,213 cases and 71,579 controls from Japanese, European, and Chinese populations. The gene-based test confirmed that the already discovered genes, *TNIP1* ($p = 3.91 \times 10^{-7}$)[7], *C9orf72* ($p = 1.96 \times 10^{-21}$)[9], *KIF5A* ($p = 1.06 \times 10^{-7}$)[6], and *SCFD1* ($p = 2.25 \times 10^{-6}$)[4]. In addition, the novel genes *ERGIC1* ($p = 5.90 \times 10^{-7}$), *RAPGEF5* ($p = 3.71 \times 10^{-7}$), *FNBP1* ($p = 9.59 \times 10^{-9}$), *ACSL5* ($p = 4.50 \times 10^{-11}$), and *ATXN3* ($p = 5.81 \times 10^{-7}$), also reached genome-wide significance (Table 2, Fig. 6a, b, Supplementary Data 5).

## Discussion

In this study, the region in *ACSL5* was discovered as a novel risk locus for sporadic ALS by meta-analysis between Japanese and European datasets and was replicated in the Chinese dataset and another Japanese dataset. Expression analysis showed that the risk allele is associated with increased *ACSL5* expression. The expression of *ACSL5* mRNA in spinal motor neurons isolated by laser-capture microdissection in 12 sporadic ALS patients and nine controls was catalogued by Batra et al.[10,11]. *ACSL5* mRNA expression was possibly higher in sporadic ALS than in controls (Supplementary Fig. 5; $p$-value $= 0.033$ with Mann–Whitney $U$ test). Similarly, another report showed that *ACSL5* mRNA expression in the spinal anterior horn was upregulated in sporadic ALS patients compared with that in controls[12].

ACSL5 is one of the members of the acyl-CoA synthetase long chain family. Acyl-CoA synthetase produces acyl-CoA for numerous metabolic pathways, such as cellular lipid metabolism; transcriptional regulation; intracellular protein transportation; and protein acylation in various tissues, including skeletal muscle, the liver, and the brain[13]. *ACSL5* is a neurotoxic A1 astrocyte-related gene, and is up-regulated in A1 astrocytes[14]. A1 astrocytes are abundant in various neurodegenerative diseases, including

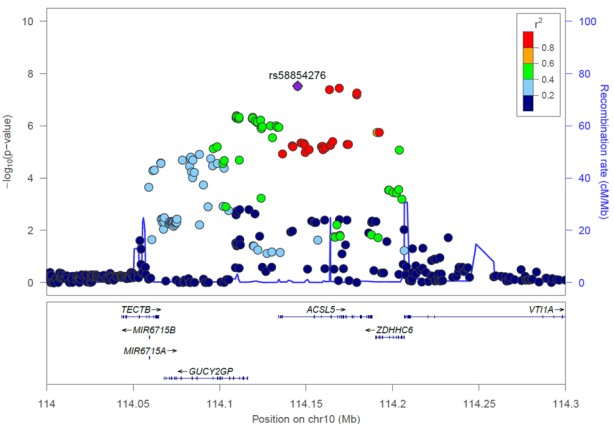

**Fig. 2 Regional ALS association plot of the *ACSL5* locus from the meta-analysis results using LocusZoom.** The meta-analysis between European and Japanese GWAS shows rs58854276 as the SNP most strongly associated with ALS ($p = 2.97 \times 10^{-8}$).

**Fig. 3 Forest plots showing the effects of rs3736947 in *ACSL5* in each cohort and meta-analysis.** Forest plots showing the effects of rs3736947 on ALS in each cohort and meta-analysis.

| Study | N(case) | N(control) | OR (95%CI) | p | |
|---|---|---|---|---|---|
| **GWAS** | | | | | |
| European | 20,806 | 59,804 | 1.07 (1.05–1.10) | 9.27e-07 | |
| Japanese | 1,173 | 8,925 | 1.16 (1.05–1.29) | 3.97e-03 | |
| **Meta-analysis (GWAS)** | 21,979 | 68,729 | 1.08 (1.06–1.09) | 3.61e-08 | |
| **Replication** | | | | | |
| Chinese | 1,234 | 2,850 | 1.20 (1.10–1.29) | 2.41e-04 | |
| Japanese 2nd | 707 | 971 | 0.96 (0.84–1.10) | 5.90e-01 | |
| **Meta-analysis (replication)** | 1,941 | 3,821 | 1.14 (1.10–1.18) | 1.97e-04 | |
| **Meta-analysis (overall)** | 23,920 | 72,550 | 1.09 (1.07–1.10) | 7.81e-11 | |

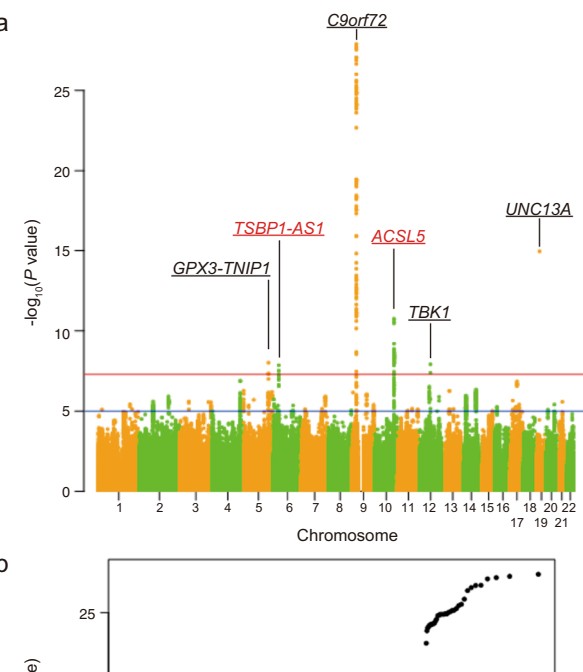

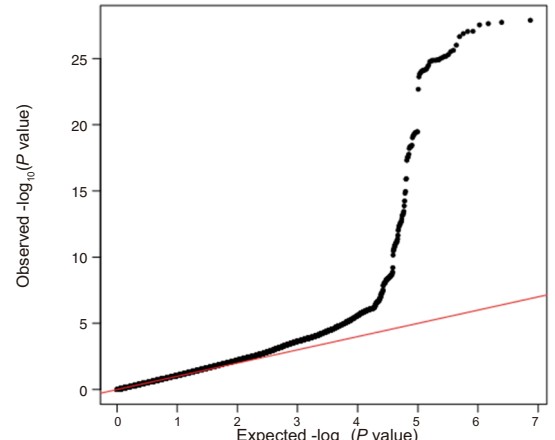

**Fig. 4 Multi-ethnic meta-analysis among European, Japanese, and Chinese genome-wide association studies (GWAS). a** Manhattan plot of the multi-ethnic meta-analysis among European, Japanese, and Chinese GWAS. The novel loci *ACSL5* and *TSBP1-AS1* are shown in red. Loci identified in other studies are presented with lines along the gene names, i.e. *GPX3-TNIP1*, *C9orf72*, *TBK1*, and *UNC13A*. The red horizontal line indicates the genome-wide significance threshold of $p = 5 \times 10^{-8}$. The blue line indicates the suggestive threshold of $p = 5 \times 10^{-6}$. The genes indicated in red text colour with lines are novel genes and other genes are known risk genes. **b** Q–Q plot of the multi-ethnic meta-analysis among European, Japanese and Chinese GWAS.

ALS, and they induce the death of neurons in the central nervous system[15]. We speculated that increased expression of *ACSL5* could induce A1 astrocytes, cause motor neuron death, and lead to ALS development. Another possible reason why *ACSL5* is associated with ALS may be related to lipid metabolism. The risk allele (A) of rs58854276 has been reported to be associated with lower HDL-cholesterol in Japanese individuals[16]. There have been several reports that dyslipidaemia increases the risk of ALS[17,18]. However, the association between ALS and dyslipidaemia has not been replicated in other studies[19]. Further studies are warranted to clarify the association between *ACSL5* and ALS onset.

In the largest meta-analysis with 23,213 cases and 71,579 controls from Japanese, European, and Chinese populations, we identified a novel locus at 6p21, which reached genome-wide significance, in addition to *ACSL5*. The top SNP in the locus (rs140736091) was in the long non-coding RNA *TSBP1-AS1*. Some studies have reported that long non-coding RNAs are associated with ALS, but their role in ALS still needs to be elucidated[20]. Therefore, further replication study and functional analysis will be needed to clarify the association between rs140736091 and patients with ALS.

The gene-based test for the largest multi-ethnic meta-GWAS from Japanese, Chinese and Europeans revealed novel genes, *ERGIC1*, *RAPGEF5*, *FNBP1*, and *ATXN3*, in addition to *ACSL5*.

*ERGIC1* is a membrane-bound protein that is localised to the endoplasmic reticulum (ER)–Golgi intermediate compartment (ERGIC). The ERGIC mediates membrane traffic and selective transport of cargo between the ER and the Golgi complex[21]. ER–Golgi transport dysfunction is reported to be a common pathogenic mechanism in *SOD1*-, *TDP-43*-, and *FUS*-associated ALS[22], and ER stress has been implicated in ALS aetiology. Combined GWAS of genetic overlap between ALS and fronto-temporal dementia-spectrum neurodegenerative diseases identified rs538622 near *ERGIC1*[23].

*RAPGEF5* is a member of the Ras subfamily of GTPases, which function in signal transduction for cell growth and differentiation as guanosine diphosphate (GDP)/guanosine triphosphate (GTP)-regulated switches cycling between inactive GDP- and active GTP-bound states[24]. *RAPGEF5* has been reported to be associated with telencephalic neurogenesis[25]. The *RAPGEF5* transcript is expressed predominantly in the brain.

FNBP1, a member of the formin-binding protein family, is a membrane-associated protein. It plays an important role in clathrin-mediated endocytosis[26]. *FNBP1* is upregulated in the spinal cord of *SOD1* G93A mice[27].

ATXN3 is a ubiquitously expressed deubiquitinase that plays important roles in the ubiquitin proteasome system, transcriptional regulation and neuroprotection[28]. A recent meta-analysis

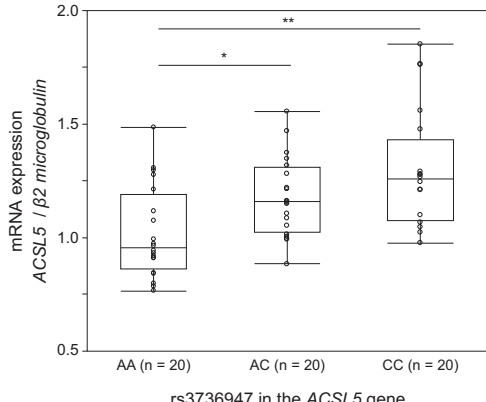

**Fig. 5 Relative expression of *ACSL5* mRNA in LCLs from ALS patients with each genotype of rs3736947 in the *ACSL5* gene.** Gene expression analysis for *ACSL5* was conducted by quantitative real-time PCR using lymphoblastoid B cell lines (LCLs) from Japanese patients with sporadic ALS. The mRNA expression of each genotype of rs3736947 was compared using the Kruskal–Wallis test followed by Steel–Dwass test. The expression of *ACSL5* was significantly higher in the LCLs of patients with the CC ($n = 20$) and AC ($n = 20$) genotype of rs3736947 than in those with the AA genotype ($n = 20$) (**CC vs AA, $p = 0.0035$, *AC vs AA, $p = 0.0166$). Circles represent individual data points. The bottom and the top of the box indicates the interquartile range (the 25th and 75th percentiles) and the line represents the median. The whiskers under and over the box correspond to the minimum and maximum values.

of European GWAS suggested an association between SNP (rs10143310) in *ATXN3* and ALS, although the SNP did not achieve genome-wide significance ($p = 3.2 \times 10^{-7}$)[6]. The CAG repeat expansion in the coding region of *ATXN3* causes spinocerebellar ataxia type 3 (SCA3)[29]. SCA3 patients and ALS patients have common pathologies, such as TDP-43-positive inclusions in the lower motor neurons of the anterior horn of the spinal cord and brainstem[30]. A gene involved in cerebellar ataxia, *ATXN2*, has also been described as a risk gene for sporadic ALS. An intermediate repeat expansion in *ATXN2* is associated with the risk of ALS[31].

In conclusion, multi-ethnic GWAS identified the association of the *ACSL5* locus with ALS. This locus was identified by combining GWAS results from our Japanese dataset with the largest set of European GWAS data and was replicated in an independent Japanese cohort and a Chinese cohort. In addition, gene-based analysis identified *ERGIC1*, *RAPGEF5*, *FNBP1*, *ACSL5*, and *ATXN3*. While these genes reached the discovery stage of the analysis, further replication analysis or functional analysis in ALS is warranted. Nevertheless, the discovery of novel risk loci significantly advances our understanding of ALS aetiology.

## Methods

**Study subjects in the Japanese cohort.** In the discovery cohort, we performed a GWAS in Japanese sporadic ALS cases from the Japanese Consortium for ALS research (JaCALS)[32] and in normal controls from the Tohoku Medical Megabank Project (TMM)[33,34]. In the replication cohort, we obtained DNA from ALS patients registered by BioBank Japan[35,36] and normal controls registered in the Pharma SNP Consortium. The ethics committees of the respective research projects approved this study. Written informed consent for this study was obtained from all the participants.

**DNA extraction from ALS patients.** To extract genomic DNA, peripheral whole-blood samples were processed using an Autopure LS system (Qiagen, Hilden, Germany) for automated nucleotide purification according to the manufacturer's instructions. We omitted RNase treatment, measured the concentration of the double-stranded DNA with PicoGreen (Life Technologies, Carlsbad, CA, USA), and adjusted the concentration of the DNA to 200 ng/μL in Elution Buffer (Qiagen).

**Table 2 Summary of significant genes in the multi-ethnic meta-analysis of gene-based association analysis among Japanese (JaCALS), European, and Chinese cohorts.**

| Ensembl Gene ID | Chr | Start | End | No of SNPS | NPARAM | N | ZSTAT | P | SYMBOL | Reported |
|---|---|---|---|---|---|---|---|---|---|---|
| ENSG00000145901 | 5 | 150409506 | 150473138 | 126 | 23 | 94792 | 4.9399 | $3.91 \times 10^{-7}$ | *TNIP1* | Benyamin, B. et al. (2017) (as *GPX3-TNIP1*)[7] |
| ENSG00000113719 | 5 | 172261278 | 172379688 | 140 | 31 | 94792 | 4.859 | $5.90 \times 10^{-7}$ | *ERGIC1* | Novel |
| ENSG00000136237 | 7 | 22157856 | 22396763 | 385 | 65 | 94792 | 4.9503 | $3.71 \times 10^{-7}$ | *RAPGEF5* | Novel |
| ENSG00000120162 | 9 | 27325207 | 27529779 | 304 | 49 | 94792 | 9.4345 | $1.96 \times 10^{-21}$ | *MOB3B* | van Es, M. A. et al. (2009) (as *C9orf72*)[9] |
| ENSG00000147896 | 9 | 27524312 | 27526496 | 5 | 2 | 94792 | 7.3632 | $8.98 \times 10^{-14}$ | *IFNK* | van Es, M. A. et al. (2009) (as *C9orf72*)[9] |
| ENSG00000147894 | 9 | 27546544 | 27573864 | 59 | 9 | 94792 | 7.399 | $6.86 \times 10^{-14}$ | *C9orf72* | van Es, M. A. et al. (2009) et al.[9] |
| ENSG00000187239 | 9 | 132649466 | 132805473 | 100 | 13 | 94792 | 5.6192 | $9.59 \times 10^{-9}$ | *FNBP1* | Novel |
| ENSG00000197142 | 10 | 114133776 | 114188138 | 49 | 7 | 94792 | 6.4829 | $4.50 \times 10^{-11}$ | *ACSL5* | Novel |
| ENSG00000023041 | 10 | 114190058 | 114206672 | 12 | 2 | 94792 | 5.1368 | $1.40 \times 10^{-7}$ | *ZDHHC6* | as *ACSL5* |
| ENSG00000135454 | 12 | 58017193 | 58027138 | 13 | 3 | 94792 | 5.1883 | $1.06 \times 10^{-7}$ | *B4GALNT1* | Nicolas, A. et al. (2018) (as *KIF5A*)[6] |
| ENSG00000092140 | 14 | 31028329 | 31089269 | 55 | 8 | 94792 | 4.944 | $3.83 \times 10^{-7}$ | *G2E3* | van Rheenen, W. et al. (2016) (as *SCFD1*)[4] |
| ENSG00000092108 | 14 | 31091318 | 31205018 | 137 | 10 | 94792 | 4.5869 | $2.25 \times 10^{-6}$ | *SCFD1* | van Rheenen, W. et al. (2016)[4] |
| ENSG00000066427 | 14 | 92524896 | 92572965 | 98 | 5 | 94792 | 4.8619 | $5.81 \times 10^{-7}$ | *ATXN3* | Novel |

*Chr* chromosome, *Start* start position of gene, *End* end position of gene. Position is based on Human Genome Assembly build 37, *No of SNPs* The number of SNPs annotated to that gene that were found in the data and were not excluded based on internal SNP QC. *NPARAM* the number of relevant parameters used in the model. *N* The sample size used when analysing that gene. *P* The gene p-value, using asymptotic sampling distribution. SYMBOL: gene name. Significant threshold of $p = 2.85 \times 10^{-6}$ (=0.05/17544).

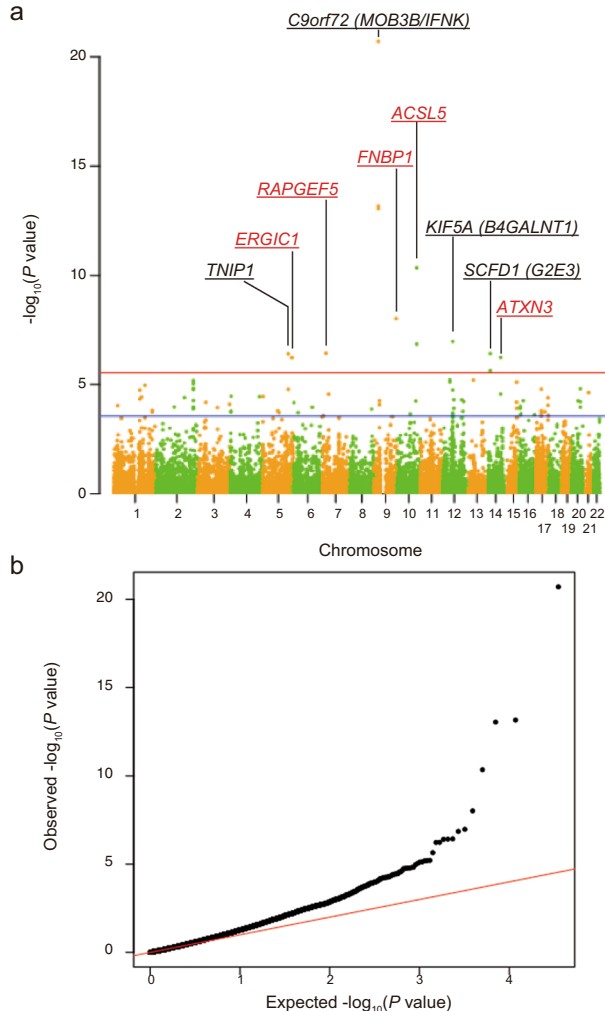

**Fig. 6 The gene-based analysis for the multi-ethnic meta-analysis among European, Japanese, and Chinese cohorts. a** Manhattan plot of the gene-based analysis for the multi-ethnic meta-analysis among European, Japanese, and Chinese cohorts. The red horizontal line indicates the genome-wide significance threshold of $p = 2.85 \times 10^{-6}$ (=0.05/17544). The blue line indicates the suggestive threshold of $p = 2.85 \times 10^{-4}$. The genes indicated in red are novel genes, and the other genes are known risk genes. **b** Q–Q plot of gene-based analysis for the multi-ethnic meta-analysis among European, Japanese, and Chinese cohorts.

**Japanese ALS patients for the discovery study.** The case cohort comprised 1,245 candidate sporadic ALS patients recruited from the Japanese Consortium for ALS research (JaCALS), which included 32 neurology facilities in Japan. Patients with a family history for ALS were excluded. The included patients were diagnosed with definite, probable, probable laboratory-supported, or possible ALS based on the revised El Escorial criteria[37] and were of Japanese ancestry. The DNA samples used in this study were collected from February 2006 to December 2017. The approval of the overall research was obtained from the Ethics Committee of Nagoya University with number 2004-0181-2. The ethics committees of all participating institutions approved the study. Details of the JaCALS have been described elsewhere[32]. All 1,245 sporadic ALS patients were genotyped with Illumina (San Diego, CA, USA) HumanOmniExpressExome-8 BeadChips (version 1.0, 1.2, and 1.3).

**Japanese controls for the discovery study.** The control cohort was the prospective cohort of the Tohoku Medical Megabank Project (TMM)[33,34]. From these TMM samples, we used 9,924 samples genotyped with Illumina HumanOmniExpressExome-8 BeadChips (version 1.2-B) at RIKEN. Approval to use the dataset as the control was obtained from the Ethical Committee of the Tohoku University Graduate School of Medicine, the TMM, and the Materials and

Information Distribution Review Committee of Tohoku Medical Megabank Organization.

**Quality control of genotyped SNPs and imputation.** For the combined genotyped dataset with 1,245 cases and 9,924 controls, we selected SNPs with minor allele frequency ≥0.03, Hardy–Weinberg equilibrium ≥0.001, and genotype call rate ≥95%. This step retained 516,447 SNPs from 936,632 SNPs. The qualified genotype dataset was imputed with the 2,037 whole-genome reference panel (2KJPN) from TMM[38] using Impute 4[39]. The target individuals themselves for the imputation were not included in the 2KJPN reference panel[40]. According to the protocol by Shido et al.[41], the imputed genotype data in Oxford GEN format were converted to Plink BED format by selecting the genotype with the highest posterior probability for each SNP and individual. In the conversion, highest posterior probabilities less than 0.9 were handled as missing genotypes. Finally, we constructed the imputed dataset with 1245 cases and 9924 controls from 27,893,192 SNPs.

**Pre-processing and GWAS.** We excluded 69 suspected ALS patients and two non-ALS patients from 1,245 cases. Samples that were identical-by-descent (>0.1875) among 1174 cases and 9924 controls were removed. The thresholds were based on other published studies[42–45]. Finally, 1173 cases and 8925 controls were used for GWAS. For GWAS, we selected 4,349,201 SNPs with minor allele frequency ≥0.03, Hardy–Weinberg equilibrium ≥0.001, and genotype call rate ≥95%. We conducted logistic regression analysis with principal components 1–20 from the principal component analysis as covariates using Plink (version 1.90b5.1).

**Meta-analysis of Japanese and European dataset.** We downloaded the summary statistics of the 20,806 ALS cases and 59,804 controls in the European dataset from the website http://als.umassmed.edu/[6]. Inverse-variance meta-analysis was conducted between the 20,806 ALS cases and 59,804 controls in the European dataset[6] and the 1173 cases and 8925 controls of Japanese ancestry using METAL (version 2011-03-25)[46]. The regions from 114,000,000 to 114,300,000 in chromosome 10, including the significant SNPs in *ACSL5*, were plotted using LocusZoom (version 1.4) with '–build hg19 –pop ASN –source 1000G_Nov2014' options[47].

**Japanese replication cohort.** We obtained DNA from 707 ALS patients registered by BioBank Japan[35,36] and 971 normal controls registered in the Pharma SNP Consortium. Twenty-nine SNPs with a suggestive threshold of $p = 5 \times 10^{-6}$ on chromosome 10q25.2 were genotyped via the multiplex PCR-based target sequencing method. Primers were designed using Primer 3 software. The products of multiplex PCR were sequenced using Illumina HiSeq 2500 (Illumina Inc., San Diego, CA, USA). The sequence data were analysed using a standard pipeline. The details of the methods have been previously described[48].

For this analysis, we did not have genome-wide genotype information for these samples. We conducted logistic regression analysis using Plink with no covariate, e.g. principal components, for these 29 SNPs.

**Meta-analysis of Chinese and Japanese replication cohort.** We downloaded the summary statistics of the 1234 ALS cases and 2850 controls in the Chinese cohort[7]. Inverse-variance meta-analysis was conducted for the 1234 ALS cases and 2850 controls in the Chinese cohort and 707 ALS cases and 971 normal controls in the Japanese replication cohort using METAL (version 2011-03-25)[46]. A value of $p < 1.72 \times 10^{-3}$ (= 0.05/29; Bonferroni adjusted) in the replication stage was considered statistically significant.

**Multi-ethnic meta-analysis.** Inverse-variance meta-analysis was conducted among 1234 cases and 2850 controls in the Chinese cohort[7] and the former samples in European[6] and Japanese datasets (JaCALS) using METAL (version 2011-03-25)[46].

**Gene-based analysis of the multi-ethnic dataset.** For the summary statistics data of the multi-ethnic meta-analysis of Japanese, European, and Chinese datasets with 3,704,464 SNPs, a gene-based analysis was applied using MAGMA (version 1.07) with the default option[49]. In the pre-processing step, each SNP was checked to be mapped to a specific gene. If the SNP was located within the gene body region of a gene, the SNP was annotated to this gene, i.e. a SNP in an intergenic region was not annotated to any gene. In the pre-processing step, an annotation window range can be set to include the peripheral regions around the genes. The default value of the annotation window range is zero in MAGMA (version 1.07). The value is the strictest option in this pre-processing step since extra regions around the gene are not included in gene-based analyses. Thus, we used the default option. After the pre-processing, 1,585,558 SNPs in total were annotated to 17,544 genes. For the gene-based analysis, a *p*-value should be calculated for each gene from the SNPs annotated to this gene in the pre-processing step. We used the default SNP-wise mean model in MAGMA for the calculation step. The SNP-wise mean model (the mean of the χ2 statistic for the SNPs in a gene) is highly similar to the commonly used SKAT model (with inverse variance weights)[50]. The drawback of this method is that it decreases the power to detect associations for rare variants. In our analysis, the minor allele frequency in the discovery Japanese cohort was more than 0.03, and the problem was negligible.

Finally, 13 genes with a genome-wide significance threshold of $p = 2.85 \times 10^{-6}$ (=0.05/17544; after Bonferroni correction) were selected.

**Quantitative real-time PCR**. Lymphoblastoid B cell lines (LCLs) were prepared from peripheral blood B cells of ALS patients using standard Epstein-Barr virus transformation techniques at the time of registration for JaCALS[51]. LCLs obtained from 20 patients with age-matched and sex-matched ALS (Supplementary Table 1) for each genotype of rs3736947 were applied to the gene expression analysis. All 60 patients whom we selected to quantify eQTL were diagnosed with sporadic ALS. Further, we performed exome sequencing or target resequencing for all 60 patients who were selected to quantify eQTL. The details of the methods have been previously described[2]. These patients had no pathogenic variant in the ALS-related genes, such as *SOD1*, *FUS*, and *TARDBP*. Total RNA was extracted from the LCLs using the PureLink RNA Mini kit (Thermo Fisher Scientific, Waltham, MA, USA). Total RNA was transcribed using the SuperScript IV VILO Master Mix (Thermo Fisher Scientific). Real-time PCR was performed using the THUNDERBIRD SYBR qPCR Mix (Toyobo, Osaka, Japan) and the CFX96 system (BioRad, Hercules, CA, USA), according to the manufacturer's instructions. The expression level of the internal control, $\beta 2$ microglobulin, was simultaneously quantified. The primers are listed in Supplementary Table 2. Differences in *ACSL5* mRNA expression among three genotype groups were evaluated by the Kruskal–Wallis test followed by the Steel–Dwass test. Statistical analyses were conducted using the JMP15.1 program (SAS Institute Inc., Cary, NC, USA).

**Statistics and reproducibility**. Software and database used for the data analysis of this study are as follows: PLINK version 1.90b5.1 (https://www.cog-genomics.org/plink2), IMPUTE4 (https://jmarchini.org/impute-4/), LocusZoom version 1.4 (https://genome.sph.umich.edu/wiki/LocusZoom_Standalone), METAL version 2011-03-25 (http://csg.sph.umich.edu/abecasis/metal/index.html), MAGMA version 1.07 (https://ctg.cncr.nl/software/magma), 2KJPN (https://ijgvd.megabank.tohoku.ac.jp/), BioBank Japan Project (https://biobankjp.org/english/index.html). Statistical analyses for mRNA expression analyses were conducted using the JMP15.1program (SAS Institute Inc., Cary, NC, USA).

**Reporting summary**. Further information on research design is available in the Nature Research Reporting Summary linked to this article.

## Data availability

The summary statistics of our genome-wide association studies are available at the Human Genetic Variation Database (Accession ID: HGV0000013). The source data of expression of *ACSL5* mRNA in LCLs are available in the Supplementary Data 4. All relevant data are available from G.S. (sobueg@aichi-med-u.ac.jp) upon request.

## Code availability

No previously unreported custom computer code was used to generate results.

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

## Acknowledgements

The authors thank all of the ALS patients who participated in this study. The authors also thank all of the participating doctors and staff of the Japanese Consortium for Amyotrophic Lateral Sclerosis research (JaCALS). The participating doctors of JaCALS are listed in the Supplementary Material. This study was mainly supported by the Japan Agency for Medical Research and Development (AMED; grant numbers JP19ek0109284h0003, JP19lk1601002h0002, JP19ak0101111h0001, JP20ek0109492h0001, JP20ak0101111h0002, and JP20dm0107059h0005). This work was also partially supported by AMED (grant numbers JP17km0105001, JP17km0105002, JP17km0105003, JP17km0105004, and JP17km0405205), and by Grants-in Aid for Scientific Research (C) KAKENHI (grant numbers JP19K06523 and JP19K07973) from the Japan Society for the Promotion of Science (JSPS). A part of the computational resources was provided by the Tohoku University Tohoku Medical Megabank Organization supercomputer system (http://sc.megabank.tohoku.ac.jp/en).

## Author contributions

R.N., Ka.M., G.T., M.Nak., M.Nag., and G.S. conceived the study and interpreted the results. R.N., Ka.M., G.T., M.Nak., S.F., and M.Nag. performed statistical and bioinformatic analyses. R.N., N.A., Na.H., D.Y., Ha.W., Hi.W., Ma.K., Y.I., K.K., No.H., M.M., A.T., O.K., M.O., K.S., S.K., N.S., M.A., Ya.O., T.Y., K.A., R.H., I.A., K.O., Ko.M., K.H., T.I., O.O., K.N., R.K., and G.S. were involved in sample collection and management of Japanese ALS cohort (JaCALS). Ka.M., S.F., N.I., N.M., and M.Nag provided Japanese GWAS control sample for the discovery cohort. Y.K., S.I., Y.M., and Mi.K. provided Japanese ALS sample and control sample for the replication cohort. Y.K., S.I., Y.M., and Mi.K. performed genotyping. G.T. and Yo.O. performed mRNA expression analysis. R.N., Ka.M., G.T., M.Nak., N.A., M.Nag., and G.S. wrote the first draft of the manuscript. All authors contributed to the revision of the manuscript.

## Competing interests

The authors declare no competing interests.

## Additional information

Ryoichi Nakamura [1,33], Kazuharu Misawa [2,3,33], Genki Tohnai[1,33], Masahiro Nakatochi [4,33], Sho Furuhashi[2], Naoki Atsuta[1], Naoki Hayashi[1], Daichi Yokoi[1,5], Hazuki Watanabe[1,6], Hirohisa Watanabe[7,8], Masahisa Katsuno [1], Yuishin Izumi[9], Kazuaki Kanai[10,11], Nobutaka Hattori[10], Mitsuya Morita[12], Akira Taniguchi[13], Osamu Kano[14], Masaya Oda[15], Kazumoto Shibuya[16], Satoshi Kuwabara[16], Naoki Suzuki [17], Masashi Aoki[17], Yasuyuki Ohta[18], Toru Yamashita [18], Koji Abe[18], Rina Hashimoto[19], Ikuko Aiba[19], Koichi Okamoto[20], Kouichi Mizoguchi[21], Kazuko Hasegawa[22], Yohei Okada[23], Tomohiko Ishihara[24], Osamu Onodera [24], Kenji Nakashima[25], Ryuji Kaji [9], Yoichiro Kamatani [26], Shiro Ikegawa [27], Yukihide Momozawa[28], Michiaki Kubo [29], Noriko Ishida[2], Naoko Minegishi[2], Masao Nagasaki [2,30,31✉] & Gen Sobue [1,7,32✉]

[1]Department of Neurology, Nagoya University Graduate School of Medicine, Nagoya, Aichi, Japan. [2]Tohoku Medical Megabank Organization, Tohoku University, Sendai, Miyagi, Japan. [3]Department of Molecular Genome Analysis, Institute of Biomedical Science, Kansai Medical University, Hirakata, Osaka, Japan. [4]Division of Data Science, Department of Nursing, Nagoya University Graduate School of Medicine, Nagoya, Aichi, Japan.

[5]Department of Neurology, Kakeyu-Misayama Rehabilitation Center Kakeyu Hospital, Ueda, Nagano, Japan. [6]Department of Neurology, Japanese Red Cross Nagoya Daiichi Hospital, Nagoya, Aichi, Japan. [7]Brain and Mind Research Center, Nagoya University, Nagoya, Aichi, Japan. [8]Department of Neurology, Fujita Health University, Toyoake, Aichi, Japan. [9]Department of Neurology, Institute of Biomedical Sciences, Tokushima University Graduate School, Tokushima, Japan. [10]Department of Neurology, Juntendo University Graduate School of Medicine, Tokyo, Japan. [11]Department of Neurology, Fukushima Medical University School of Medicine, Fukushima, Japan. [12]Division of Neurology, Department of Internal Medicine, Jichi Medical University, Shimotsuke, Tochigi, Japan. [13]Department of Neurology, Mie University Graduate School of Medicine, Tsu, Mie, Japan. [14]Division of Neurology, Department of Internal Medicine, Toho University Faculty of Medicine, Tokyo, Japan. [15]Department of Neurology, Vihara Hananosato Hospital, Miyoshi, Hiroshima, Japan. [16]Department of Neurology, Graduate School of Medicine, Chiba University, Chiba, Japan. [17]Department of Neurology, Tohoku University School of Medicine, Sendai, Miyagi, Japan. [18]Department of Neurology, Okayama University Graduate School of Medicine, Okayama, Japan. [19]Department of Neurology, National Hospital Organization Higashinagoya National Hospital, Nagoya, Aichi, Japan. [20]Department of Neurology, Geriatrics Research Institute and Hospital, Maebashi, Gunma, Japan. [21]Department of Neurology, National Hospital Organization Shizuoka Medical Center, Shizuoka, Japan. [22]Division of Neurology, National Hospital Organization, Sagamihara National Hospital, Sagamihara, Kanagawa, Japan. [23]Department of Neurology, Aichi Medical University, Nagakute, Aichi, Japan. [24]Department of Neurology, Brain Research Institute, Niigata University, Niigata, Japan. [25]Department of Neurology, National Hospital Organization, Matsue Medical Center, Matsue, Shimane, Japan. [26]Laboratory for Statistical and Translational Genetics, RIKEN Center for Integrative Medical Sciences, Yokohama, Kanagawa, Japan. [27]Laboratory for Bone and Joint Diseases, RIKEN Center for Integrative Medical Sciences, Tokyo, Japan. [28]Laboratory for Genotyping Development, RIKEN Center for Integrative Medical Sciences, Kanagawa, Japan. [29]RIKEN Center for Integrative Medical Sciences, Yokohama, Kanagawa, Japan. [30]Center for the Promotion of Interdisciplinary Education and Research, Kyoto University, Sakyo-ku, Kyoto, Japan. [31]Center for Genomic Medicine, Kyoto University Graduate School of Medicine, Sakyo-ku, Kyoto, Japan. [32]Aichi Medical University, Nagakute, Aichi, Japan. [33]These authors contributed equally: Ryoichi Nakamura, Kazuharu Misawa, Genki Tohnai, Masahiro Nakatochi. ✉email: nagasaki@genome.med.kyoto-u.ac.jp; sobueg@aichi-med-u.ac.jp

