## [Peer Review File · Communications Biology]

Reviewers' comments:

Reviewer #1 (Remarks to the Author):

This is a genome-wide analysis of a small cohort of Japanese ALS patients and a meta-analysis with a larger European cohort.

There are a number of significant issues with this paper that make the conclusions outlined in this manuscript preliminary:

- The discovery cohort of Japanese patients was small and likely underpowered.
- The ACSL5 notably did not replicate in the second replication cohort of Japanese. The replication signal is driven by the Chinese cohort. This is concerning.
- The replication cohort does not seem to have been corrected for population stratification.
- There are technical issues with respect to how the GWAS analysis was performed that need to be addressed:

o Why was a minor allele frequency of 0.03 chosen? The standard cutoffs are 0.05 and 0.01. I don't think I've seen that before. Please provide a reference to justify this.

o Why was a PIHAT cutoff of 0.1875 used? The standard cutoff is either 0.1 or 0.125 (corresponding to third degree relatives). Inclusion of related individuals can lead to inflation of p-values.

o What was the R2 threshold applied to imputed SNPs? Furthermore, the typical genotype call rate for a GWAS is 95%. Please provide a reference to justify the lower cutoff threshold. What was the specific R2 and genotype call rates of the SNPs evaluated in the replication cohort?

Minor

- The atxn3 locus was reported in the Nicolas et al, Neuron 2019 paper.

Reviewer #2 (Remarks to the Author):

Nakamura and colleagues have performed a GWAS meta-analysis of ALS cases and controls of Japanese descent, European descent and Chinese descent. They identify a new association at ACSL5 and find that the top SNP is an eQTL for ACSL5. They also find altered expression of ACSL5 in a small number of ALS cases compared to controls.

Overall, the manuscript is an important one to get out to the ALS community as it helps in diversifying the patients studied in ALS genetics and identifies what appears to be a robust association at ACSL5. There are various aspects of the work that I think could be elaborated on a bit more, and I have detailed them below. I also think the narrative could use a bit more restructuring. It feels a bit at the moment that it hops around (to association results, replication, some functional information, then back to associations again) and might benefit from a more streamlined telling of the story. Please see below for all my comments.

Major comments

[1] On page 8, line 124, you say the associated SNPs are in the gene body region. What are their functional annotations? Are they missense, nonsense, intronic, etc? Are they in high linkage disequilibrium for any regulator SNPs, eQTLs, etc? It looks like you've found a SNP that is an eQTL for ACSL5, but I'm curious if there are other SNPs in that LD block that have other regulatory information? Or is ACSL5 truly the only SNP that has a mechanistic link to the top association you're seeing?

[2] Are there other mechanistic links at the other loci you mention? (ERGIC1, etc?) I would be curious to know if the authors have any intuition around what other causal genes might be?

[3] Why did you select LCLs to test for ACSL5 expression? Was this the only tissue available to you? There are perhaps other more disease-relevant cells that might have been more informative (e.g., motor neurons)? But I realize that this tissue may not have been available to the authors. Also, in this test, you only show people with the AA or CC genotype. Were there no people who were AC to test? That would likely be a helpful additional group of individuals, to see if there is indeed a true correlation. In fact, is there a way to perform a formal colocalization analysis to test if rs3736947 is the association signal for both the GWAS signal and the eQTL?

[4] What was the genetic background of the cells selected to quantify ACSL5 expression? Do they have familial mutations or are they all sporadic cases?

[5] Could you please provide a bit more detail on the gene-based tests in the main body of the text? Did you use a frequency cut-off for these? Did you use particular functional annotation classes (nonsynonymous, loss-of-function, etc)? What sort of test did you use (SKAT, T1, etc). I know you mention using MAGMA in the methods, but adding a bit more detail as to what, precisely, that means would be very helpful to readers who are unaware.

Minor comments

[1] In the abstract, you mention running an association study in a Japanese case/control set but then (line 87) say you run a combined meta-analysis of individuals with European ancestry. I assume you mean a meta-analysis of your Japanese samples and additional European samples? If so, clarifying the sentence to say " a combined meta-analysis of our Japanese cohort with samples of European descent" (or something along those lines) would be helpful.

[2] Line 99 seems to suggest that ALS ends in death or permanent dependence on a respirator. But to my knowledge, all ALS cases (with the exception of perhaps Stephen Hawking, who has a very different form of ALS) end in death. Please clarify.

[3] Accompanying QQ plots with your Manhattan plots would be helpful, to get a sense of what the overall distribution of the test statistics looks like (I see you have them in the supplement but they could be insets in the main figures?). A forest plot for the ACSL5 SNP would also be helpful, as it would allow readers to see how consistent the direction of effects are. This would also allow readers to see what the joint analysis across all studies looks like.

[4] The narrative can feel a bit jumpy; there are comments about replicating certain SNPs in separate cohorts, discuss an eQTL finding, and then discuss running a full meta-analysis at the end anyway (line 147, which will be the better powered analysis). I might consider shuffling the narrative just a bit so that the GWAS / meta-analysis comes first, followed by any additional follow-up of the loci (eQTLs, colocalizations, etc). I think it will help the reader understand a bit better.

[5] I would caution the authors to check what they mean when they significant. For example, I would argue given the many tests performed here that the Batra et al data (line 161) does not show significance at $p = 0.033$ once you adjust for multiple testing. I think a bit more cautious wording there would be good. Also, is this data available by genotype or only by phenotype? I.e., could we see expression in cases and controls stratified by genotype? How do we know that expression is linked to genotype and phenotype.

We are grateful to all the reviewers for their excellent suggestions regarding our initial submission. We found their comments to be extremely valuable and helpful, and we have addressed each point raised. Our responses to the reviewers' comments are as follows.

Reviewer #1:

This is a genome-wide analysis of a small cohort of Japanese ALS patients and a meta-analysis with a larger European cohort.

There are a number of significant issues with this paper that make the conclusions outlined in this manuscript preliminary:

1) The discovery cohort of Japanese patients was small and likely underpowered.

In our study, the discovery was not achieved solely from a Japanese cohort. The discovery cohort is the result of a meta-analysis with our Japanese cohort and a European cohort study. By combining these cohorts, we could select the candidate SNPs related to ALS. To confirm this result, we applied a combined replication study with another Japanese cohort and Chinese cohort.

2) The ACSL5 notably did not replicate in the second replication cohort of Japanese. The replication signal is driven by the Chinese cohort. This is concerning.

The Japanese replication result does not show the result to be controversial to the result of the discovery study.

The data in Table 1 show that the replication study solely from the Japanese cohort cannot conclude that rs3736947 has a significant effect to cause ALS. Thus, we have included additional replication data from a Chinese cohort and combined the Chinese and Japanese cohorts. Accordingly, we could conclude that rs3736947 has a significant effect on ALS. Statistically, the effect size of rs3736947 in ACSL5 is low; thus, the result cannot be conclusive in one study. For better clarity, we have also added forest plots in Figure 3.

3) The replication cohort does not seem to have been corrected for population stratification.

Thank you very much for bringing this point to our attention. Yes, we did not include principal components in the replication analysis. Unfortunately, we do not have genome-wide genotyping information for these samples. We applied the target sequencing-based approach for the replication study to the 29 candidate SNPs, as described in the Materials and Methods section.

We have added the following description in the Materials and Methods section (page 20, line 324-326) to clarify it.

‘For this analysis, we do not have genome-wide genotype information for these samples. We conducted logistic regression analysis using Plink with no covariate, e.g. principal components, for these 29 SNPs.’

There are technical issues with respect to how the GWAS analysis was performed that need to be addressed:

4) Why was a minor allele frequency of 0.03 chosen? The standard cutoffs are 0.05 and 0.01. I don’t think I’ve seen that before. Please provide a reference to justify this.

For example, the following paper uses $MAF \geq 0.03$ for GWAS: Wossenie et al (2019) ‘Genome wide association study of body weight and feed efficiency traits in a commercial broiler chicken population, a re-visitation’, *Scientific Reports*.

<https://www.nature.com/articles/s41598-018-37216-z>.

5) Why was a PIHAT cutoff of 0.1875 used? The standard cutoff is either 0.1 or 0.125 (corresponding to third degree relatives). Inclusion of related individuals can lead to inflation of p-values.

We used the 0.1875 cutoff (half-way point between 2nd and 3rd degree relatives) on the basis of some published studies [Ellington et al, Wuttke et al, and ILAECCS; mentioned below]. In addition, to reduce population stratification, we considered 20 principal components.

The combination of these parameters sometimes underpowers the GWAS result; therefore, the use of these parameters in this analysis would address the inflation problem. The relevant references have been included in the revised manuscript. (page 18, line 301). Please note the following:

According to Anderson et al. (2010) ‘Data Quality Control in Genetic Case-Control Association Studies’ in *Nature Protocol*, the threshold 0.1875 is one of the cut-offs in GWAS.

According to Ellingson et al. (2016) ‘Automated quality control for genome wide association studies’ in *F1000Res*, the threshold 0.1875 represents the half-way point between 2nd and 3rd degree relatives and is a common cut-off to use.

The International League Against Epilepsy Consortium on Complex Epilepsies (2018) ‘Genome-wide mega-analysis identifies 16 loci and highlights diverse biological mechanisms in the common epilepsies’ in *Nature Communications* removed one member from each sample pair with $IBD > 0.1875$, with the exception

of the EPGP familial epilepsy cohort.

According to Wuttke et al. (2019) ‘A catalog of genetic loci associated with kidney function from analyses of a million individuals’ in *Nature Genetics*, only one member of each pair with an IBD coefficient ≥ 0.1875 was retained.

6) What was the R2 threshold applied to imputed SNPs? Furthermore, the typical genotype call rate for a GWAS is 95%. Please provide a reference to justify the lower cutoff threshold. What was the specific R2 and genotype call rates of the SNPs evaluated in the replication cohort?

We apologize for the error in the description of the QC pass threshold for the genotyping SNP call rate. In fact, the QC pass threshold of the call rate of genotyping SNPs was 95% in our analysis. The threshold of call rate of imputed genotyping SNPs was also 95% in our analysis. The former manuscript was not correct; we have updated this in the revised manuscript (page 19, line 304).

After the QC SNPs, we applied whole-genome imputation with a Japanese reference panel with very similar steps as in Shido et al (2019). After whole-genome imputation was applied to the QC genotyping dataset using impute4, the imputed genotype data in Oxford GEN format were converted to Plink BED format by selecting the genotype with the highest posterior probability for each SNP and individual. In the conversion, highest posterior probabilities of <0.9 were handled as missing genotypes. This conversion is conservative approach compared to the using all imputed SNP with a very low info score. The threshold of 95% call rate for the imputed SNP means ‘95% individuals should have very reliable info score for the imputed genotyping result.’

To be clear, the following sentences have been added to the main text (pages 18, lines 291–295).

‘According to the protocol in Shido et al.³⁷, the imputed genotype data in Oxford GEN format were converted to Plink BED format by selecting the genotype with the highest posterior probability for each SNP and individual. In the conversion, highest posterior probabilities less than 0.9 were handled as missing genotypes.’

In addition, we have added the information of imputation quality scores (info scores) of imputed SNPs and the call rates of genotyping SNPs (direct typing SNP) to Table 1 (pages 28 and 29). Two near-lead SNPs, rs58854276 and rs11195948, are imputed SNPs, with info scores of 0.995 and 0.999, respectively. The other lead SNP rs3736947 is a direct genotyped SNP, with a call rate of 1.0.

All three SNPs in the Japanese replication cohort were the results of direct genotyping using direct sequencing with multiplex PCR. The call rates of rs58854276, rs11195948, and rs3736947 were 0.995, 0.996, and 0.997, respectively.

The Chinese cohort was taken from Benyamin et al. (2013), and the replicated SNP rs3736947 was a directly genotyped SNP in the Illumina HumanOmni ZhongHua-8 array. The genotyping call rate was more than 0.99.

rs11195948 is an imputed SNP using the 1000 Genomes Project Phase 1 v3, with imputation quality score > 0.8 and HWE $p > 10^{-6}$ from their paper.

Minor

7) The atxn3 locus was reported in the Nicolas et al, Neuron 2019 paper.

We have included the relevant reference and information in the Discussion section (page15, line237-240).

Reviewer #2:

Nakamura and colleagues have performed a GWAS meta-analysis of ALS cases and controls of Japanese descent, European descent and Chinese descent. They identify a new association at *ACSL5* and find that the top SNP is an eQTL for *ACSL5*. They also find altered expression of *ACSL5* in a small number of ALS cases compared to controls.

Overall, the manuscript is an important one to get out to the ALS community as it helps in diversifying the patients studied in ALS genetics and identifies what appears to be a robust association at *ACSL5*. There are various aspects of the work that I think could be elaborated on a bit more, and I have detailed them below. I also think the narrative could use a bit more restructuring. It feels a bit at the moment that it hops around (to association results, replication, some functional information, then back to associations again) and might benefit from a more streamlined telling of the story. Please see below for all my comments.

Major comments

[1] On page 8, line 124, you say the associated SNPs are in the gene body region. What are their functional annotations? Are they missense, nonsense, intronic, etc? Are they in high linkage disequilibrium for any regulator SNPs, eQTLs, etc? It looks like you've found a SNP that is an eQTL for *ACSL5*, but I'm curious if there are other SNPs in that LD block that have other regulatory information? Or is *ACSL5* truly the only SNP that has a mechanistic link to the top association you're seeing?

Thank you for your comment. The top three SNPs (rs58854276, rs11195948, rs3736947) in *ACSL5* are all intronic variants. We obtained eQTL data of peripheral blood cells of 98 Japanese individuals from Jenger (<http://jenger.riken.jp/>). The table below shows that the top three SNPs were significantly associated with eQTL of *ACSL5* over the other genes located around the three SNPs. As mentioned in the Discussion, *ACSL5* mRNA expression in spinal motor neurons is higher in sporadic ALS than in controls (Supplementary Figure 4, Batra et al.). Similarly, another report showed that *ACSL5* mRNA expression in the spinal anterior horn was upregulated in sporadic ALS patients than in controls (Andres-Benito et al., 2017, Aging). Therefore, we considered the top three SNPs to be associated with the expression of *ACSL5* and the risk of ALS. Moreover, we checked SNPs in *ACSL5*, which reached $p < 5 \times 10^{-7}$ in the discovery stage on Regulome DB, and rs37369467 was retained as it seemed more likely to have regulatory effects (rank 3a, score 0.29611) as predicted on RegulomeDB. Since the top SNP (rs37369467) is the most significant in RegulomeDB, we thought it might be a causal variant. However, further experiments are needed to identify causal

variant.

Table. eQTL for peripheral blood of the top three SNPs in 98 Japanese normal controls from Jenger

Target	SNP	Ch	POS	REF	ALT	beta	tstat	p.value	gene	hgnc_symbol
Peripheral_blood.txt.gz	rs58854276	10	114145044	A	G	-0.79762	-7.37469	5.84E-11	ENSG00000197142.6	ACSL5
Peripheral_blood.txt.gz	rs3736947	10	114169190	C	A	-0.72903	-6.25944	1.08E-08	ENSG00000197142.6	ACSL5
Peripheral_blood.txt.gz	rs11195948	10	114163515	C	T	-0.72984	-6.29928	9.01E-09	ENSG00000197142.6	ACSL5
Peripheral_blood.txt.gz	rs3736947	10	114169190	C	A	-0.10368	-0.7524	0.453653	ENSG00000119927.9	GPAM
Peripheral_blood.txt.gz	rs58854276	10	114145044	A	G	-0.10547	-0.78159	0.436378	ENSG00000119927.9	GPAM
Peripheral_blood.txt.gz	rs11195948	10	114163515	C	T	-0.11481	-0.83657	0.404909	ENSG00000119927.9	GPAM
Peripheral_blood.txt.gz	rs11195948	10	114163515	C	T	0.013895	0.100874	0.919861	ENSG00000148737.11	TCF7L2
Peripheral_blood.txt.gz	rs58854276	10	114145044	A	G	0.014719	0.108736	0.913639	ENSG00000148737.11	TCF7L2
Peripheral_blood.txt.gz	rs3736947	10	114169190	C	A	0.014659	0.106062	0.915755	ENSG00000148737.11	TCF7L2
Peripheral_blood.txt.gz	rs3736947	10	114169190	C	A	0.221185	1.622074	0.108067	ENSG00000151532.9	VTIIA
Peripheral_blood.txt.gz	rs58854276	10	114145044	A	G	0.152551	1.134386	0.259457	ENSG00000151532.9	VTIIA
Peripheral_blood.txt.gz	rs11195948	10	114163515	C	T	0.217416	1.599238	0.113054	ENSG00000151532.9	VTIIA
Peripheral_blood.txt.gz	rs3736947	10	114169190	C	A	0.267952	1.977747	0.050824	ENSG0000023041.7	ZDHHC6
Peripheral_blood.txt.gz	rs11195948	10	114163515	C	T	0.272185	2.017398	0.046446	ENSG0000023041.7	ZDHHC6
Peripheral_blood.txt.gz	rs58854276	10	114145044	A	G	0.246857	1.855895	0.066536	ENSG0000023041.7	ZDHHC6

[2] Are there other mechanistic links at the other loci you mention? (ERGIC1, etc?) I would be curious to know if the authors have any intuition around what other causal genes might be?

We have described the details of our gene-based test in the answer to your question in the revised main text (page21-22, line 342-359). For each gene, we used the SNPs in the gene body region of the gene and did not use any SNPs outside the gene body region. This option is the strictest option for the gene-based analysis in MAGMA and would lower the risk of detecting another gene near the causal gene.

In addition, as in Supplementary Table 4, the novel genes that reached statistical significance did not have any neighboring genes that had near-equal p-values to those of the lead genes. Thus, we selected these lead genes as novel genes in our analysis.

[3] Why did you select LCLs to test for ACSL5 expression? Was this the only tissue available to you? There are perhaps other more disease-relevant cells that might have been more informative (e.g., motor neurons)? But I realize that this tissue may not have been available to the authors.

Thank you for providing these insights. Unfortunately, we could get only lymphoblastoid B cell lines (LCLs) from Japanese ALS patients. We could not get autopsy tissue or iPSC-derived motor neurons from ALS patients whose SNP genotypes in *ACSL5* were sequenced.

Also, in this test, you only show people with the AA or CC genotype. Were there no people who were AC to test? That would likely be a helpful additional group of individuals, to see if there is indeed a true correlation. In fact, is there a way to perform a formal colocalization analysis to test if rs3736947 is the association signal for both the GWAS signal and the eQTL?

In line with this suggestion, we reconduted gene expression analysis for *ACSL5*, by quantitative real-time PCR using lymphoblastoid B cell lines (LCLs) from Japanese sporadic ALS patients with CC, AC, and AA genotypes of rs3736947. We have revised Figure 5 and the sentence (page 11, lines 167–172).

Based on the data of GTEx, Japanese eQTL data of peripheral blood cells, and RNAseq data from spinal motor neurons (Batra et al.) and RegulomeDB, as we answered in response to your question [1], we considered the SNPs in *ACSL5* as the most likely causes.

Figure 5. Relative expression of *ACSL5* mRNA in LCLs from ALS patients with each genotype of rs3736947 in the *ACSL5* gene.

[4] What was the genetic background of the cells selected to quantify ACSL5 expression? Do they have familial mutations or are they all sporadic cases?

Thank you for your comment. All patients whom we selected to quantify eQTL were diagnosed with sporadic ALS. We performed exome sequencing or target resequencing for all patients who were selected to quantify eQTL using the method we previously reported (Nakamura et al. 2016, *Neurobiol Aging*) and they had no pathogenic variant in the ALS-related genes. We added the sentence below in the Methods section (page 22, line 364-370).

‘All patients whom we selected to quantify eQTL were diagnosed with sporadic ALS. We performed exome sequencing or target resequencing for all 60 patients who were selected to quantify eQTL. The details of the methods were previously described ², and they had no pathogenic variant in the ALS-related genes, such as *SOD1*, *FUS*, and *TARDBP*.’

[5] Could you please provide a bit more detail on the gene-based tests in the main body of the text? Did you use a frequency cut-off for these? Did you use particular functional annotation classes (nonsynonymous, loss-of-function, etc)? What sort of test did you use (SKAT, T1, etc). I know you mention using MAGMA in the methods, but adding a bit more detail as to what, precisely, that means would be very helpful to readers who are unaware.

Thank you for the insightful comment. We have extensively updated the main text in ‘Gene-based analysis of the meta-analysis of Japanese, European, and Chinese datasets’ to include the points that you specified as follows (pages 21–22, lines 342–359):

‘For the summary statistics data of the multi-ethnic meta-analysis of Japanese, European, and Chinese datasets with 3,704,464 SNPs, a gene-based analysis was applied using MAGMA (version 1.07) with the default option. In the pre-processing step, each SNP was checked to be mapped to a specific gene. If the SNP is located within the gene body region of a gene, the SNP is annotated to this gene, i.e. SNPs in the intergenic region are not annotated to any gene. In the pre-processing step, an annotation window range can be set to include the peripheral regions around the genes. The default value of the annotation window range is zero in MAGMA (version 1.07). This value is the strictest option in this pre-processing step since the extra region around the gene is not included in the gene-based analysis. Thus, we used the default option. After the pre-processing, 1,585,558 SNPs in total were annotated to 17,544 genes. For the gene-based analysis, a p-value should be calculated for each gene from the SNPs annotated to this gene in the pre-processing step. We used the default SNP-wise mean model in MAGMA for the calculation step. The SNP-wise mean model (the mean of the χ^2 statistic for the SNPs in a gene) is highly similar to the commonly used SKAT model

(with inverse variance weights)⁴⁸. The drawback of this method is that it decreases the power to detect associations for rare variants. In our analysis, the minor allele frequency in the discovery Japanese cohort was more than 0.03, and the problem was negligible. Finally, 13 genes with a genome-wide significance threshold of $p = 2.85 \times 10^{-6}$ ($= 0.05/17544$; after Bonferroni correction) were selected.’

Minor comments

[1] In the abstract, you mention running an association study in a Japanese case/control set but then (line 87) say you run a combined meta-analysis of individuals with European ancestry. I assume you mean a meta-analysis of your Japanese samples and additional European samples? If so, clarifying the sentence to say “ a combined meta-analysis of our Japanese cohort with samples of European descent” (or something along those lines) would be helpful.

We have modified this sentence as suggested (page 6, lines 87–88).

[2] Line 99 seems to suggest that ALS ends in death or permanent dependence on a respirator. But too my knowledge, all ALS cases (with the exception of perhaps Stephen Hawking, who has a very different form of ALS) end in death. Please clarify.

Thank you for your suggestion. About 20–30% of Japanese patients with ALS undergo tracheostomy invasive ventilation (TIV) treatment, and our previous study showed that Japanese ALS patients with TIV had longer survival times than those without TIV, with a difference of about 7 years in median survival (Hayashi et al. (2020) J Neurol Neurosurg Psychiatry doi: 10.1136/jnnp-2019-322213). Therefore, we used the phrase ‘leads to permanent dependence on a respirator within 3–5 years from disease onset.’ However, we agree with your comment, and we have revised the sentence (page 7, lines 97–99) because the text was confusing.

[3] Accompanying QQ plots with your Manhattan plots would be helpful, to get a sense of what the overall distribution of the test statistics looks like (I see you have them in the supplement but they could be insets in the main figures?). A forest plot for the ACSL5 SNP would also be helpful, as it would allow readers to see how consistent the direction of effects are. This would also allow readers to see what the joint analysis across all studies looks like.

We agree with you. We have added QQ plots with Manhattan plots in Figures 1, 4, and 6, and we have added forest plots in Figure 3.

Figure 3. Forest plots showing the effects of top three SNPs in ACSL5 in each cohort and meta-analysis.

[4] The narrative can feel a bit jumpy; there are comments about replicating certain SNPs in separate cohorts, discuss an eQTL finding, and then discuss running a full meta-analysis at the end anyway (line 147, which will be the better powered analysis). I might consider shuffling the narrative just a bit so that the GWAS / meta-analysis comes first, followed by any additional follow-up of the loci (eQTLs, colocalizations, etc). I think it will help the reader understand a bit better.

Thank you for your suggestion. We have revised the manuscript as recommended. We have divided the Results section with short subheadings, changing GWAS/multi-ethnic meta-analysis first, followed by functional analysis such as eQTL of *ACSL5*, and finally, the gene-based test.

We added the results of full meta-analysis (European, Japanese, and Chinese GWAS) (page 10, line 152-159). In the multi-ethnic meta-analysis among European, Japanese, and Chinese GWAS, the SNPs in *ACSL5* also reached genome-wide significance, and novel locus in novel locus in the long non-coding RNA *TSBP1-AS1* reached the genome-wide significance (rs140736091, $p = 1.36 \times 10^{-8}$). We also added Manhattan plot of multi-ethnic meta-analysis among European, Japanese, and Chinese GWAS, and QQ plot in the Figure 4.

Figure 4. Multi-ethnic meta-analysis among European, Japanese, and Chinese genome-wide association studies (GWAS).

a Manhattan plot of the multi-ethnic meta-analysis among European, Japanese, and Chinese GWAS.

b Q-Q plot of the multi-ethnic meta-analysis among European, Japanese and Chinese GWAS.

[5] I would caution the authors to check what they mean when they significant. For example, I would argue given the many tests performed here that the Batra et al data (line 161) does not show significance at $p = 0.033$ once you adjust for multiple testing. I think a bit more cautious wording there would be good. Also, is this data available by genotype or only by phenotype? I.e., could we see expression in cases and controls stratified by genotype? How do we know that expression is linked to genotype and phenotype.

Thank you for your suggestion. As pointed out, we have changed the words 'significantly higher' to 'upregulated' in the Discussion section (page12, line 192-194). We have also changed the words 'higher' to 'possibly higher' in the Discussion section (page12, line 191).

Batra et al. performed laser capture microdissection (LCM) of spinal motor neurons from 13 sporadic ALS patients and 9 control patients, followed by RNAseq analysis. We obtained the RNA seq data from the NCBI Gene Expression Omnibus (accession number GSE76220). Unfortunately, we could obtain only their phenotype data, not their genotype data.

REVIEWERS' COMMENTS:

Reviewer #1 (Remarks to the Author):

The authors have responded adequately to my comments and those of the other reviewers. The manuscript is improved as a consequence.

Reviewer #2 (Remarks to the Author):

The authors have been incredibly rigorous in their responses to both reviewers. The text has been updated quite substantially and additional analyses - to address questions from reviewers - have been added.

I think the appropriate caution has been included where necessary (e.g., noting that the SNP does not seem to trend in the same direction in the second Japanese replication cohort, but the confidence interval here is very large as the sample is quite small). In addition, the plots the authors have added (forest plots for index SNPs, and correlation between SNP and ACSL5 expression) are all very useful.

All in all, I think the manuscript has improved substantially and I have no further comments.

We are grateful to all the reviewers for their excellent suggestions regarding our second submission. Our point-by-point responses to each of the reviewers' comments are provided below.

Reviewer #1:

The authors have responded adequately to my comments and those of the other reviewers. The manuscript is improved as a consequence.

Thank you for your comment. We believe that our study makes a significant contribution to the genetic architecture of ALS.

Reviewer #2:

The authors have been incredibly rigorous in their responses to both reviewers. The text has been updated quite substantially and additional analyses - to address questions from reviewers - have been added.

I think the appropriate caution has been included where necessary (e.g., noting that the SNP does not seem to trend in the same direction in the second Japanese replication cohort, but the confidence interval here is very large as the sample is quite small). In addition, the plots the authors have added (forest plots for index SNPs, and correlation between SNP and ACSL5 expression) are all very useful.

All in all, I think the manuscript has improved substantially and I have no further comments.

Thank you for your comment. We believe that our study makes a significant contribution to the genetic architecture of ALS.